# Clinical Application of Dynamic SPECT/CT in a Patient with Prior Myocardial Infarction Underwent Percutaneous Coronary Intervention Twice

**DOI:** 10.3390/diagnostics11061028

**Published:** 2021-06-03

**Authors:** Hung-Pin Chan, Chin Hu, Ming-Hui Yang, Hung-Yen Chan, Nan-Jing Peng

**Affiliations:** 1Department of Nuclear Medicine, Kaohsiung Veterans General Hospital, Kaohsiung 813414, Taiwan; markscience05@hotmail.com (H.-P.C.); ghu@vghks.gov.tw (C.H.); hongyenchan0407@yahoo.com.tw (H.-Y.C.); 2Department of Medical Education and Research, Kaohsiung Veterans General Hospital, Kaohsiung 813414, Taiwan; mhyang@vghks.gov.tw; 3Department of Nuclear Medicine, Taipei Veterans General Hospital, Taipei 11217, Taiwan; 4School of Medicine, National Yang-Ming University, Taipei 11221, Taiwan

**Keywords:** coronary artery disease, dynamic SPECT/CT, myocardial infarction, myocardial perfusion imaging, percutaneous coronary intervention

## Abstract

We present a case of CAD with anteroseptal MI after stent insertion for revascularization due to symptoms presented. MPI with dynamic SPECT/CT provided useful information in terms of flow parameters and matched territories of stenting results as well as providing coronary artery flow phenomenon underwent PCI. In this case, dynamic SPECT/CT may minimize errors with proper stents treatment, especially for controversial MPI results.

A Myocardial Infarction Patient’s Imaging in Dynamic SPECT/CT. 

**Figure 1 diagnostics-11-01028-f001:**
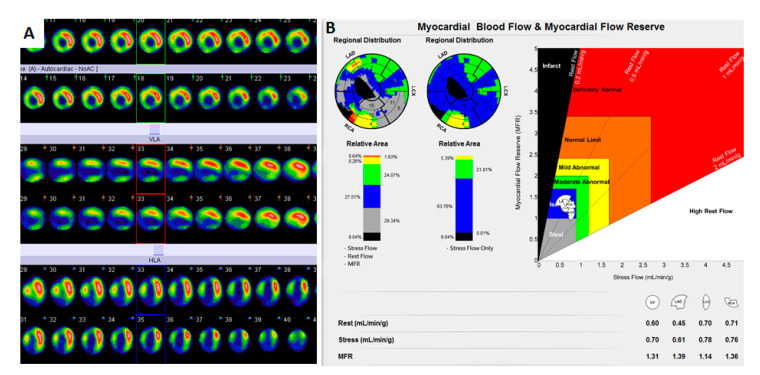
A 77-year-old man experienced coronary artery disease (CAD) with anteroseptal myocardial infarction (MI) and received a percutaneous coronary intervention (PCI) over the left anterior descending coronary artery (LAD) and right coronary artery (RCA) territories. Six years after, he complained of progressive dyspnea and chest tightness on exertion. He was referred for Tc-99m sestamibi myocardial perfusion imaging (MPI) with dynamic single-photon emission computerized tomography/computerized tomography (dynamic SPECT/CT) by a cardiologist. It was performed using a one-day rest/dipyridamole-stress protocol on a dedicated Siemens Symbia-T2 SPECT system. At rest, 13 mCi MIBI was intravenous injection with low dose CT for attenuation correlation [1]. Three hours later, dipyridamole (0.56 mg/kg) was intravenously injected, and the second dose of 30 mCi MIBI injection later. MyoFloQ software was used for data analyses [2]. Conventional MPI revealed persistent defects in the apex, anteroseptal, septum, inferior, and inferolateral walls (**A**). Dynamic SPECT/CT demonstrated 9.04% infarction in LAD territory coexisted with 63.76% ischemic myocardia (**B**).

**Figure 2 diagnostics-11-01028-f002:**
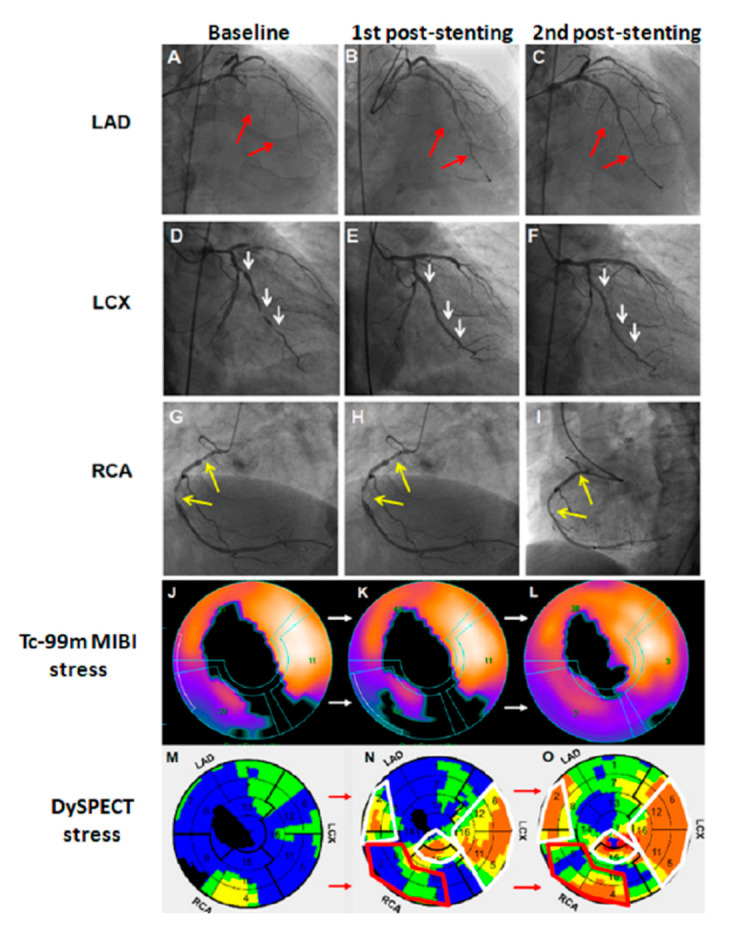
Due to the 63.76% extent of the ischemic myocardia seen on Dynamic SPECT/CT, he underwent PCI intervention, which revealed 80% stenosis over the left main (LM), total occlusion over LAD (**A**, red arrows), 90% stenosis over the left circumflex artery (LCX) (**D**, white arrows), and in-stent restenosis over the RCA (**G**, yellow arrows). To rescue the jeopardized myocardium, stents were inserted over the LAD to the LM (**B**, red arrows), LCX (**E**, white arrows) but not over the RCA orifice (**H**, yellow arrows). Two months later, Dynamic SPECT/CT was arranged again for post-PCI follow-up. The stress extent of MPI revealed similar defects (**J**,**K**) with a 2.41% decreased infarction area in the LAD territory and remarkably increased stress myocardial blood flow (MBF) on dynamic SPECT/CT (**M**,**N**; areas outlined in white). Secondary PCI was performed one month later, and stents were inserted over RCA territory (**I**, yellow arrows). LAD and LCx territories showed no in-stent restenosis (**C,F**; red and white arrows. The following MPI demonstrated partial improvement in RCA and LCX territories, but a defect remained in the LAD territory (**L**). Dynamic SPECT/CT revealed improved flow in the RCA territory (**N**,**O**, area outlined in red).

Interestingly, the further improvement in MBF to LAD and LCX territories (**N** and **O**, areas outlined in white) corresponding to increased blood flow in adjunctive branches (**C** and **F**), without any more stenting in the second PCI. The overall ischemic myocardia dropped from 63.76% to 15.24%, and the infarction area disappeared in the Dynamic SPECT/CT images. After the second PCI, this patient reported remission of cardiac symptoms and kept risk factors controlled. Quantitative MBF by cardiac PET improves diagnostic accuracy and yields better resolution [3]. Nevertheless, cardiac PET imaging has limitations due to high cost and sometimes low cyclotron availability. Hsu et al. reported the use of dynamic SPECT/CT to measure absolute MBF to enhance CAD detection and diagnosis [2,4]. According to this case, we found that dynamic SPECT/CT could provide much earlier information in terms of flow parameters to rescue the jeopardized myocardium in post-PCI evaluation. This can be a useful tool for cardiologists having patients with MI who underwent PCI. It also provides coronary artery flow phenomenon before and after PCI interventions and may provide more precise results and proper stents treatment, especially for controversial MPI results. Overall, this patient’s symptoms were improved after treatment. 

## Data Availability

Not applicable.

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
