# Peer review of "Clinical Application of Dynamic SPECT/CT in a Patient with Prior Myocardial Infarction Underwent Percutaneous Coronary Intervention Twice"

_diagnostics, 2021, doi:10.3390/diagnostics11061028_

Round 1
Reviewer 1 Report
Thank you for the opportunity to review this manuscript. The manuscript addresses an important topic. However, I recommend to take the following suggestions into consideration.
Major suggestions:
Page 1: " Due to ongoing complaints of progressive dyspnea and chest tightness on exertion": How long after the first PCI did this symptoms occur? Did they correlate with ECG changes or an elevated Troponin T? For which reason was not another coronarangiograpy performed?
Minor Suggestions:
Page 2: Please replace "interesting" by "interestingly" or another term.
"After PCI": Please specify if to was the second PCI after which symptoms improved.
"This can be a useful tool for cardiologists having patients with MI underwent PCI".: Is meant caring for these patients?
"It also provided coronary artery flow phenomenon underwent PCI": What is meant by underwent?
Author Response
Dear reviewer:
Thank you for your comments. We replied in attached file.
Dr Peng Nan-Jing

Reviewer 2 Report
It is a bit unclear as to the order of events. What was the time between the MI and the first SPECT/CT? What happened after the first procedure? Was this an acute in stent thrombosis in the LAD? Was this procedure related, or medication adherence? The angiogram after the first PCI suggeests a very poor result.
It is unlikely that a specialised test such as SPECT/CT is needed to diagnose an occluded LAD with severe left main disease. History and a more available stress test (eg stress echo) would have detected a significant abnormality.
Is there really anything clinically useful demonstrated by the SPECT/CT?
Author Response

(The authors gave the same response as above.)

Reviewer 3 Report
Dear authors,
I think that this case report is interesting and the topic is surely of clinical interest. The comparison of compared the SPECT/CT MFR results to clinical findings and to invasive coronary angiography is a hot topic. However, in the paper there are some points to improve to let the paper more clear and complete.
After these improvement and corrections, I think that the article could be accepted.
COMMENTS
The main issue of this paper is the absence of the acquisition parameters and system data.
Is it a SPECT/CT CZT-based system or a conventional dual-head single photon emission computed tomography?
As described in a recent editorial (https://doi.org/10.1007/s12350-020-02297-9) the lack of standardization is one of the main issues of this technique. The authors have obtained very interesting results in this single case. The acquisition protocol, the processing parameters and the kinetic model should be described to improve the readers knowledge.
Furthermore, since CT was used, if these parameters were in agreement with the recent best practices for CT-based attenuation correction (AC) in nuclear cardiology, please cite:
- Camoni, L., Santos, A., Attard, M. et al. Best practice for the nuclear medicine technologist in CT-based attenuation correction and calcium score for nuclear cardiology. European J Hybrid Imaging 4, 11 (2020). https://doi.org/10.1186/s41824-020-00080-0
“Hsu et al. reported the use of Dynamic SPECT/CT to measure absolute MBF to enhance CAD detection and diagnosis”. The cited study used a conventional dual-head single photon emission computed tomography gamma camera, as the one of authors, I can suppose.
If instead, the system used is a CZT-based, due to the use of the CT, please cite also:
- Giubbini, R., Bertoli, M., Durmo, R. et al. Comparison between N13NH3-PET and 99mTc-Tetrofosmin-CZT SPECT in the evaluation of absolute myocardial blood flow and flow reserve. J. Nucl. Cardiol. (2019). https://doi.org/10.1007/s12350-019-01939-x
- Zavadovsky KV, Mochula AV, Boshchenko AA, et al. 468 Absolute myocardial blood flows derived by dynamic CZT scan vs invasive fractional flow reserve: Correlation and accuracy. J Nucl Cardiol (2019). Epub 2019/03/09. doi:10.1007/s12350-019-01678-470z
Finally, the Figure 1B, must be uploaded using an higher resolution. It’s hard to read the values.
Author Response

(The authors gave the same response as above.)
